# Pressure-Enhanced Photocurrent in One-Dimensional SbSI via Lone-Pair Electron Reconfiguration

**DOI:** 10.3390/ma15113845

**Published:** 2022-05-27

**Authors:** Tianbiao Liu, Kejun Bu, Qian Zhang, Peijie Zhang, Songhao Guo, Jiayuan Liang, Bihan Wang, Haiyan Zheng, Yonggang Wang, Wenge Yang, Xujie Lü

**Affiliations:** 1Center for High Pressure Science and Technology Advanced Research, Shanghai 201203, China; tianbiao.liu@hpstar.ac.cn (T.L.); qian.zhang@hpstar.ac.cn (Q.Z.); peijie.zhang@hpstar.ac.cn (P.Z.); songhao.guo@hpstar.ac.cn (S.G.); jiayuan.liang@hpstar.ac.cn (J.L.); bihan.wang@hpstar.ac.cn (B.W.); haiyan.zheng@hpstar.ac.cn (H.Z.); yonggang.wang@hpstar.ac.cn (Y.W.); yangwg@hpstar.ac.cn (W.Y.); 2College of Material and Chemical Engineering, Zhongyuan University of Technology, Zhengzhou 450007, China

**Keywords:** high pressure, lone-pair electrons, ferroelectric semiconductor, photocurrent

## Abstract

Understanding the relationships between the local structures and physical properties of low-dimensional ferroelectrics is of both fundamental and practical importance. Here, pressure-induced enhancement in the photocurrent of SbSI is observed by using pressure to regulate the lone-pair electrons (LPEs). The reconfiguration of LPEs under pressure leads to the inversion symmetry broken in the crystal structure and an optimum bandgap according to the Shockley–Queisser limit. The increased polarization caused by the stereochemical expression of LPEs results in a significantly enhanced photocurrent at 14 GPa. Our research enriches the foundational understanding of structure–property relationships by regulating the stereochemical role of LPEs and offers a distinctive approach to the design of ferroelectric-photovoltaic materials.

## 1. Introduction

Ferroelectric materials with intrinsic spontaneous polarization have shown great promise for applications in advanced sensors, random-access memory devices, and photovoltaics [1,2,3,4]. Although significant progress has been made, ferroelectrics with appropriate bandgaps in the visible range and high photovoltaic performances are still scarce [1,5]. Recently, symmetry-breaking sulfides, such as Ba_3_Zr_2_S_7_ [6], SbSI, and CuInP_2_S_6_ [7,8,9,10], contain the less electronegative sulfur atom and have smaller bandgaps compared with the ferroelectric oxides, which have been developed as potential ferroelectric materials [11].

The behavior of lone-pair electrons (LPEs) in the ferroelectric sulfides containing 14- and 15-group metal ions has gained increasing research interest due to the enhanced polarization from the stereochemical expression of LPEs [12,13,14,15]. The ns^2^ lone pairs (n is the group number) have shown an exceptionally rich chemistry because of the combination of stereochemical activity and structural flexibility [16,17,18,19]. Such behavior of LPEs results in unusual anharmonic lattice dynamics for polar distortions linked to the special optoelectronic properties [16,17]. Particularly, the 5s^2^-Sb^3+^ in one-dimensional ferroelectric semiconductor SbSI drives ion displacement and leads to ferroelectric phase transition at a Curie temperature near room temperature (T_C_ = 295 K) [20]. The strong lattice polarization is significant for the generation of large shift currents, which can considerably improve the efficiency of charge separation and migration [7,21]. Therefore, the large polarization and suitable bandgap make SbSI a promising candidate for solar cell applications [22]. Despite the great importance of LPEs in SbSI, the dynamic behavior of LPEs remains unclear. Addressing the challenge requires advanced in situ characterization methods.

As an alternative thermodynamic parameter, pressure can effectively modify the local structure and coordination, and further achieve modification of LPEs [23,24,25,26,27,28,29]. In this work, by using high pressure to modulate the local structure of SbSI, the stereochemical role of LPEs is controllably manipulated towards designable optoelectronic properties. The increased polarization driven by LPEs is crucial for the improvement of charge separation and migration, resulting in an enhanced photocurrent. Our research provides new insights into the optimization of optoelectronic properties by regulating the stereochemical role of LPEs ferroelectric semiconductors.

## 2. Materials and Methods

### 2.1. Sample Preparation

The SbSI crystal was synthesized by the chemical vapor deposition method (CVD). Sb_2_S_3_ and SbI_3_ powders were used as solid precursors.
Sb_2_S_3_ + SbI_3_---------3SbSI(1)

We put the starting materials Sb_2_S_3_ and SbI_3_ into the center and upper stream of the furnace, respectively. The furnace was pumped to 0.5 Torr and maintained the pressure at 100 Torr by argon filled. Then, the furnace was heated rapidly to 650 K and reached a higher melting point for samples. The furnace was kept heating for 40 min and finally cooled down naturally. Finally, the dark red SbSI crystals were obtained. After cooling, the samples were rinsed in ethanol to remove residual SbI_3_. The morphology of obtained crystals was investigated by a scanning electron microscope (SEM) (JEOL, JSM6510). The composition of single crystals was examined by energy-dispersive X-ray spectroscopy (EDXS) (Oxford). As shown in Appendix A, the SEM image presents large sizes of SbSI crystals (>100 μm). The presence of Sb, S, and I are confirmed by semiquantitative EDXS as shown in Appendix A. The Sb/S/I ratio is approximately 1:1:1 in Appendix A, which is in accordance with the stoichiometric ratio of SbSI.

### 2.2. In Situ High-Pressure Characterizations

The symmetric Diamond Anvil Cell (DAC) was used in high-pressure experiments, where the size of the diamond anvils was 300 μm. We used laser-drilling to pre-indent a stainless-steel gasket to 150 μm diameter and 50μm thickness hole, which served as the sample chamber. The SbSI crystals and a small ruby ball were placed into the sample chamber. The ruby fluorescence method was used to determine the pressures [30]. In our in situ high-pressure characterizations, optical absorption and X-ray diffraction (XRD) experiments used silicone oil as a pressure transmitting medium [31], while no pressure-transmitting medium was used in resistance and photocurrent measurement. All the high-pressure measurements are at room temperature of 300 K.

### 2.3. XRD Measurements

In situ high-pressure powder XRD experiments were conducted at 4W2 station of the Beijing Synchrotron Radiation Facility. The wavelength of the X-ray beam is 0.6199 Å. The CeO_2_ was used to calibrate the distance and tilting of the detector. In experiments, we use the FIT2D program to collect, integrate and deal with 2D XRD images. Structure refinements were carried out by using the Rietveld method in JANA2006 software [32]. Fitting cell volume data to the third-order Birch–Murnaghan equation of state:(2)$$P\left(V\right)=\frac{{3B}_{0}}{2}\left[{\left(\frac{V_{0}}{V}\right)}^{\frac{7}{3}}-{\left(\frac{V_{0}}{V}\right)}^{\frac{5}{3}}\right]\left\{1+\frac{3}{4}\left(B\prime -4\right)\left[{\left(\frac{V_{0}}{V}\right)}^{\frac{2}{3}}-1\right]\right\}\;$$
where *V_0_* is the initial volume, *V* is the deformed volume, and *B′* is the derivative of the bulk modulus with respect to pressure [33].

### 2.4. Optical Absorption Measurements

We chose the 60 × 15 μm size of rodlike shape single-crystal SbSI and loaded it into the DAC chamber for the high-pressure optical absorption measurements. The absorption spectra and optical images were measured in a micro-region spectroscopy system (Gora-UVN-FL, built by Ideaoptics, Shanghai, China). The absorption spectra were carried out between 300 and 1000 nm by using a Xenon light source. The fiber spectrometer is a Nova, ideaoptics spectrometer. A camera (MiChrome 5 Pro, Hinckley, UK) equipped with a microscope was used to record the photographs of the sample under the same conditions including exposure time and intensity. The bandgaps at different pressures are assessed by extrapolating the linear portion of the *α*^1/2^ versus the *hν* curve, where *α* is the absorption coefficient and *hν* is the photon energy [34].

### 2.5. Resistance and Photocurrent Measurement

The crystal was carefully grounded before loading into the DAC chamber for high-pressure electrical experiments. In situ resistance was measured by a van de Pauw (vdP) method. The current source, nano voltmeter, and voltage/current switch system were used the Keithley 6221, 2182A, and 7001m, respectively [35]. In order to avoid a short circuit, the cubic boron nitride was coated on a steel gasket, which served as an electrical insulator. For the photocurrent measurement, a two point-probe method was conducted under a 20 W incandescent lamp.

### 2.6. Density Functional Theory (DFT) Calculations

The DFT calculation was performed by the Vienna ab initio simulation package (VASP) [36]. The Perdew-Burke-Ernzerhof (PBE) of the general gradient approximation (GGA) function was employed in the present work [37]. The projector augmented wave (PAW) potential was used on a plane-wave basis. Here, a plane wave cut-off energy was 400 eV. A 3 ×3 × 3 k-mesh was used for the conventional cell in structure optimization. To understand the bonding character of SbSI during compression, the electron localization function (ELF) was calculated [38,39]:(3)$$\mathrm{ELF}\left(r\right)=\{1+[\frac{K\left(r\right)}{\mathrm{Kh}\left(\rho \left(r\right)\right)}{]}^{2}{\}}^{-1}$$
where *ρ*(r) is the density at r, *K* is the curvature of the electron pair density, and *Kh*(*ρ*(r)) is the value of *K* under density *ρ* homogeneous electron gas.

## 3. Results and Discussion

The crystal structure of the paraelectric SbSI with the *Pnma* space group consists of double chains of covalent bonds formed along the *c*-axis at ambient conditions. The weak electrostatic interactions exist in the perpendicular direction, which originated from the interlayer coupling of LPEs (Figure 1). As shown in Figure 1a, the in situ structural evolution of SbSI up to 28.5 GPa was investigated by synchrotron XRD. The selected Rietveld analysis XRD profiles are shown in Appendix A and Appendix A. During compression, all diffraction peaks shifted to higher angles monotonically, indicating the shrinkage of the lattice and the volume compression. It is worth noting that the paraelectric to ferroelectric phase transition occurs due to the symmetry break at 2.9 GPa, evidenced by the variation of diffraction peaks in XRD. Rietveld refinements of the XRD patterns at 2.9 GPa confirmed that the space group from *Pnma* to *Pna*2_1_ is caused by the off-centering displacement of LPEs. Furthermore, the isostructural transition from one-dimensional to three-dimensional (1D to 3D) occurs around 15 GPa, which mainly arises from the suppression of the LPEs (Figure 1a) [16]. The evolutions of lattice parameters of SbSI under pressure are shown in Figure 1b. The *a* and *b* directions were more compressible than the *c* direction, which contributes to the shrinkage of the interchain. The variation of the cell volumes of SbSI during compression is displayed in Figure 1c. The values of bulk modulus (*B_0_*) 36.5(2) GPa for the 1D structure and 63.4(8) GPa for the 3D structure were obtained by fitting the Birch–Murnaghan equation of state, which indicates the less compressible nature due to the formation of Sb−I rigid bonding [16]. The decompression process indicates that the phase transition is reversible (Appendix A).

The optical absorption spectroscopy measurements of SbSI up to 30.0 GPa were conducted to examine the bandgap evolution. As shown in Figure 2a, the absorption edge was found at about 650 nm (1.89 eV), which is in line with the previous report. The redshift of the absorption edge is ascribed to the shortening of bonds, which leads to the broadening of the valence and conduction bands. During compression, the variation of bandgap and slope of band edge show a significant change at 14.2 GPa (Figure 2b and Appendix A), which indicates a phase transition. Such behavior is consistent with 1D to 3D structural transition around 15 GPa. According to Shockley–Queisser theory, an optimum bandgap is about 1.34 eV [40,41]. With increasing pressure, the variation of the bandgap for SbSI suggests that an optimum bandgap of 1.34 eV is achieved at around 15 GPa (Figure 3b). Thus, considering the optimized bandgap, it is presumable to expect enhanced optoelectronic properties.

The photocurrents of SbSI were measured with an on–off switch at different bias voltages up to 23 GPa. It is reported that the ferroelectrics have an intrinsic internal built-in electric field that separates photo-generated charge carriers and thus have a photocurrent at zero bias voltage [22,42]. SbSI initially had no zero-bias photoresponse, which is consistent with the paraelectric property. The pressure-induced obvious photocurrent was observed from 2.6 to 14 GPa at zero bias voltage (Figure 3a), which indicates the emergence of the ferroelectric feature under high pressure. With increasing pressure, the photocurrent significantly enhances and then reaches the maximum value at 14 GPa at different bias voltages (Figure 3b,c and Appendix A). The stereochemical role of LPEs brings a highly distorted local structure at 14 GPa, which gives rise to enhanced spontaneous polarization. Such behavior considerably improves the efficiency of charge separation and migration relied on enhanced spontaneous polarization, which results in the enhancement of photocurrents under pressure. This measurement deepens the fundamental understanding of the relationship between LPEs and ferroelectric-photovoltaic properties, which will guide the exploration of new devices with emergent properties by ambient methods [43].

The resistivity evolution of SbSI as a function of pressure up to 25 GPa was measured by the vdP method (Figure 3d) [35]. The resistivity increases during compression up to 12 GPa. The abnormally increasing resistivity is attributed to the suppression of the LPEs, which leads to distorted local structures with increased electron scattering and thus hinders electron transport. Upon further compression, the decrease in resistivity is associated with the disappearance of LPEs, which results in the 1D to 3D isostructural transition and enhances electron transport.

To simulate the stereochemical expression of LPEs in SbSI under high pressures, we carried out density functional theory (DFT) calculations on the band structure and electron localization function (ELF) [38,39]. The band structure of SbSI has strong dispersions along the S-Y points, while the Γ-Z-U-R-T-X points are relatively flat, which are typical for electronically 1D band structural motifs (Appendix A) [44,45]. As shown in Figure 4, it is seen that an apparent non-spherical lobe charge distributes from the Sb atoms, which is regarded as the Sb^3+^-5s^2^ LPEs [46]. The Sb^3+^-5s^2^ LPEs are gradually suppressed and eventually disappeared above 14.8 GPa, which indicates the disappearance of electrostatic interactions between chains and the formation of Sb-I bonding (Figure 1 and Figure 4). Furthermore, the interchain between Sb and I atoms shows low charge distributions in the valence electron density map at 1.2 GPa, which indicates weak electrostatic interactions (Figure 4a). The apparent distribution of charge along the Sb-I direction suggests a strong covalent bond at 28.5 GPa, resulting in the formation of 3D structures (Figure 4b). Therefore, our experimental results and theoretical simulations provide consistent findings which provide new insights into the structure-relationship of ferroelectric semiconductors from the perspective of lone-pair behavior.

## 4. Conclusions

In conclusion, we have revealed the relationship between crystal structures and optoelectronic properties of SbSI under high pressures by in situ experimental characterization and DFT calculations, including XRD, absorption spectroscopy, photocurrent, and resistivity measurements. The reconfiguration of LPEs causes structural symmetry-breaking and an optimum bandgap according to the Shockley–Queisser limit. Furthermore, by regulating stereochemical LPEs, SbSI shows an enhanced photocurrent under high pressures. This work provides new insights into the fundamental understanding of the stereochemical expression of LPEs in ferroelectric photovoltaics.

## Figures and Tables

**Figure 1 materials-15-03845-f001:**
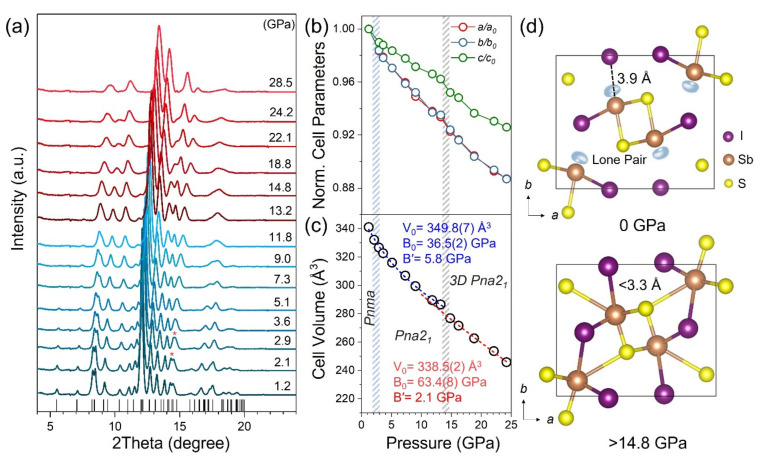
**Pressure-induced structural evolution of SbSI.** (**a**) Comparison of SbSI XRD at different pressures. (**b**) The lattice constants fitting by the Rietveld method and (**c**) the unit-cell volume as a function of pressure. (**d**) The crystal structures at 0 and 22.8 GPa.

**Figure 2 materials-15-03845-f002:**
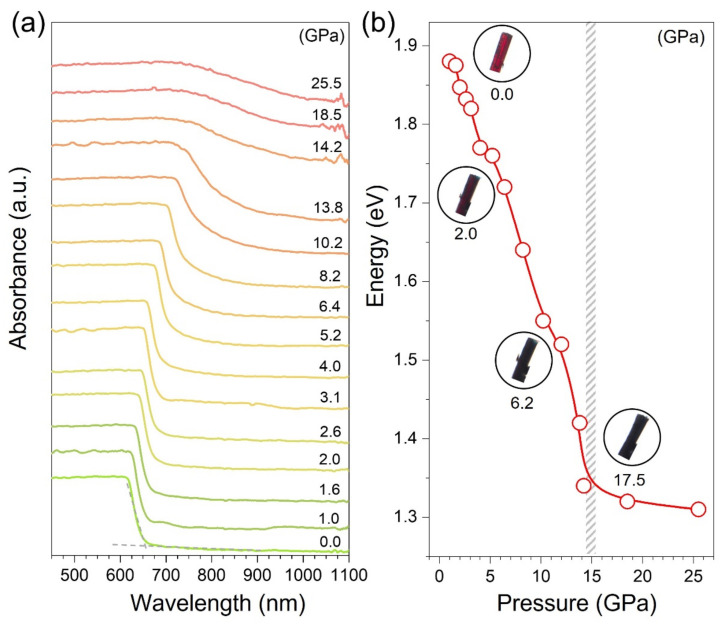
**Optical absorption spectra of SbSI under pressure.** (**a**) Optical absorption spectra of SbSI under pressure. (**b**) Pressure dependence of the bandgap energy for SbSI. Inset: optical micrographs of SbSI collected at different pressures.

**Figure 3 materials-15-03845-f003:**
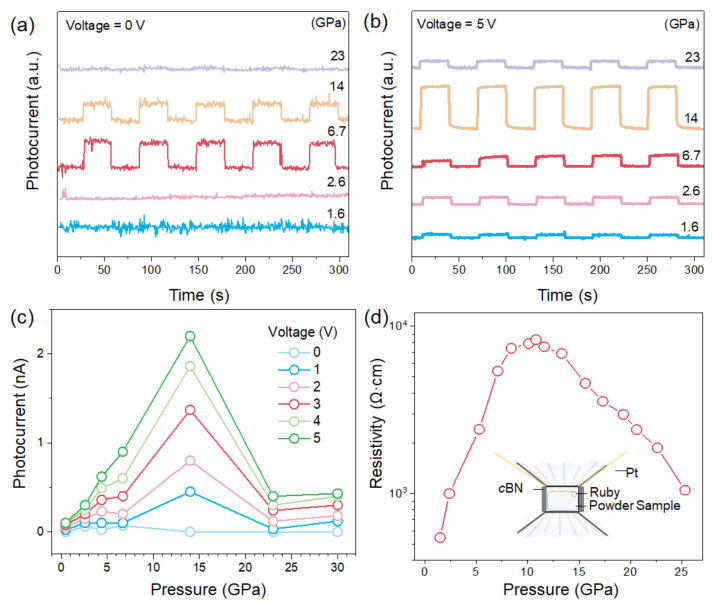
**Photocurrent and resistivity of SbSI at different pressures.** (**a**,**b**) Pressure-dependent photocurrent of SbSI at 0 and 5 V bias voltages. (**c**) Photocurrent as a function of pressure at different bias voltages. (**d**) The resistivity of SbSI under high pressures. The inset shows the schematic diagram for resistivity measurement.

**Figure 4 materials-15-03845-f004:**
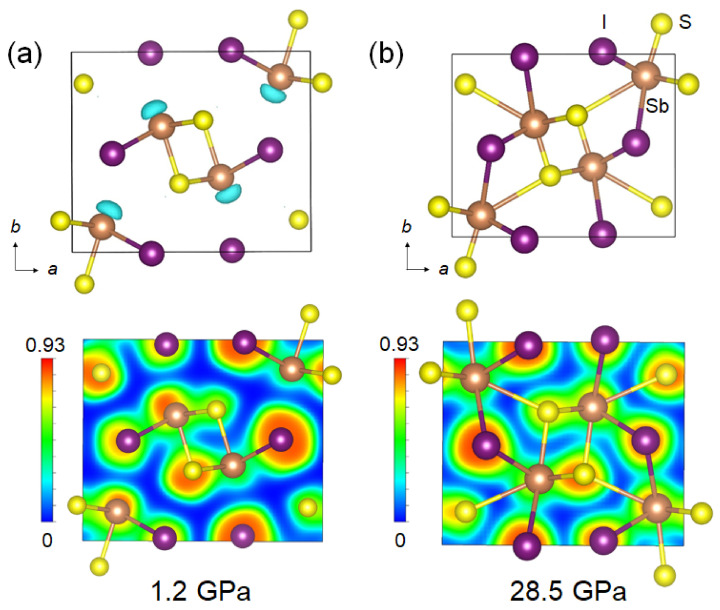
**Electron localization function (ELF) of SbSI at different pressures.** The calculated ELF plots and ELF in the (0 0 1) sections of SbSI at (**a**) 1.2 GPa and (**b**) 28.5 GPa. The isosurface value ranges from 0 to 0.93.

## Data Availability

The data presented in this study are available on request from the corresponding author.

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
