# Peer review of "Pressure-Enhanced Photocurrent in One-Dimensional SbSI via Lone-Pair Electron Reconfiguration"

_materials, 2022, doi:10.3390/ma15113845_

Round 1

Reviewer 1 Report

The whole topic and the idea about SbSI becoming ferroelectric under P and revealing photocurrent is interesting. However, I find multiple problems in reporting of structural and optical properties, which prevents me from recommending publishing this work. These drawbacks are confusing and can create wrong impressions for the readership.

  1. structural investigation. I do not trust the results of the Rietveld refinements, especially at high pressures. The ticks do not correspond to the observed peak positions, the quality figures of the fit are bad. I do not believe that the positions of the atoms (not presented but shown in a figure) save the behavior of the lone pairs can be determined from these data. I believe that only single-crystal XRD could address the structural behavior in this case.

Minor comments to the presentation: the color coding in Fig. 1(a) is unclear, the atoms in the Fig. 3d are not labeled. The fits to the Volume vs P curves are not presented adequately, e.g. the derivative of the bulk modulus is not presented as well as the uncertainties. Presenting the EOS of an inferred high-pressure phase using the value of the parameters at 0 GPa is very confusing and yield the results, which are difficult to interpret. If the high-pressure phase has an extended structure (opposite to a chain structure at 0 GPa), it should be less compressible and have smaller initial volume. The opposite has been reported, which makes no sense.

  1. optical absorption. It is unclear how the spectra could be measured using a visualization camera and microscope.

The data presented in a.u. are confusing. It is unclear how the band gaps were determined and what kind of absorption edge it is assumed to deduce this. From visual inspection it is clear that the electronic structure changes drastically between 8.2 and 14.2 GPa which is different from the claimed 15 GPa.

The notion of the optimum bandgap of 1.34 eV and how was it determined is unclear.   

Reviewer 3 Report

The paper looks interesting.  I suggest that it could be accepted after some revision, and here are my suggestions for the authors.

In my opinion, the Authors should explain why they use the expression "Quasi-one-dimensional " and their conclusions about the 1D-to-3D transition.

The abbreviation DAC should be explained 

The size of obtained sample should be specified. There is no research confirming that the resulting material is SbSI

In Chapter 2 the Authors wrote: “resistance was measured by a four-point-probe resistance” and in Chapter 3 they wrote: “The resistivity evolution… was measured by four-probe Van de Pauw method”. The Authors should specify the exact method of measurement. The van der Pauw method is used for the measurement of surface resistivity –they should comment on this fact. What is the influence of strong anisotropy of SbSI on the resistivity measurements?

The Authors should explain the presence of photocurrent at zero bias voltage

The authors should add information about the temperature of the sample during the experiment.

Because the Curie temperature is a very important parameter of ferroelectric material, in my opinion, the Authors should complete their research with pressure dependency of it.

Round 2

Reviewer 1 Report

I recommend publishing as the revision is sufficiently good

Reviewer 2 Report

The reviewer thanks the authors for considering the requested comments.

Reviewer 3 Report

SbSI is a 3D crystal and in this context, the authors should rather justify the use of the quasi-one-dimensional term than change it into one-dimension. I think such an explanation should be introduced in the text.

I think the information about the dimensions of both obtained and chosen to examine crystal should be introduced in the text.

I think that the text should include information on how the sample was adapted to electrical measurements.

I think that title of the paper is a bit misleading because of two reasons: 1 in my opinion SbSI crystals are not one-dimensional crystals, (the previous version "quasi-one-dimensional" was more proper on the condition it would be cleared up in the text) 2. during the investigation the SbSI was not in ferroelectric phase (300K> Tc) suggesting by the title - it also should be clearly mentioned in the text.
